# Nurture Early for Optimal Nutrition (NEON) programme: qualitative study of drivers of infant feeding and care practices in a British-Bangladeshi population

Monica Lakhanpaul ,[1,2] Lorna Benton,[1] Oliver Lloyd-Houldey,[1] Logan Manikam,[3,4] Diana Margot Rosenthal,[1] Shereen Allaham,[3,4] Michelle Heys[1,5]

ML and LB contributed equally.

ML and LB are joint first authors.

For numbered affiliations see end of article.

**Correspondence to**
Professor Monica Lakhanpaul; m.lakhanpaul@ucl.ac.uk

## ABSTRACT

**Objectives** To explore optimal infant feeding and care practices and their drivers within the British-Bangladeshi population of East London, UK, as an exemplar to inform development of a tailored, coadapted participatory community intervention.

**Design** Qualitative community-based participatory research.

**Setting** Community and children's centres and National Health Service settings within Tower Hamlets, London, UK.

**Participants** 141 participants completed the community study including: British-Bangladeshi mothers, fathers, grandmothers and grandfathers of infants and young children aged 6–23 months, key informants and lay community members from the British-Bangladeshi population of Tower Hamlets, and health professionals working in Tower Hamlets.

**Results** 141 participants from all settings and generations identified several infant feeding and care practices and wider socioecological factors that could be targeted to optimise nutritional outcomes. Our modifiable infant feeding and care practices were highlighted: untimely introduction of semi and solid foods, overfeeding, prolonged parent-led feeding and feeding to 'fill the belly'. Wider socioecological determinants were highlighted, categorised here as: (1) society and culture (e.g. equating 'chubby baby' to healthy baby), (2) physical and local environment (e.g. fast food outlets, advertising) and (3) information and awareness (e.g. communication with healthcare professionals around cultural norms).

**Conclusions** Parenting interventions should be codeveloped with communities and tailored to recognise and take account of social and cultural norms and influence from different generations that inform infant feeding and care practices and may be of particular importance for infants from ethnically diverse communities. In addition, UK infant feeding environment requires better regulation of marketing of foods for infants and young children if it is to optimise nutrition in the early years.

### Strengths and limitations of this study

► Existing literature does not adequately describe the cultural and social determinants of infant feeding and care practices. This study addresses this gap by generating transferable lessons from the British-Bangladeshi population to inform similar studies to inform codeveloped, adapted or tailored programmes sensitive to the needs of culturally diverse communities.

► A strength of this study is the development of culturally and generational sensitive interview tools and recruitment strategies. The methodology for the development and the tools/strategies themselves can be generalised to other populations.

► A further strength of the study that the data collected directly informed the wider Nurture Early for Optimal Nutrition intervention led by the same study group thereby ensuring rapid transfer of results into a public health intervention.

► The strong partnership developed between community members from the British-Bangladeshi population of Tower Hamlets, London, UK, enabled a wider reach and diversity of representation in this study. The study engaged not only with mothers but also members from the wider community and across generations.

► One limitation of this study and potential for future research is that the qualitative data were not triangulated with a quantitative measure of dietary intake (out of scope for this study). Another limitation is that all data are self-reported and have the potential for respondent bias. However, we report data from a large sample with thematic saturation.

## INTRODUCTION

The first 1000 days of life presents a critical window, but also an opportunity to prevent the dual burden of undernutrition and overnutrition. In the UK, an estimated 5% of the population is undernourished, of which 70% goes unreported.[1] Meanwhile,

obesity prevalence continues to rise in accordance with projections that over half of UK adults could be obese by 2050.[2] Currently, approximately one-third of children leave primary school as obese or overweight in the UK.[3] Obesity, alongside potential mediating and wider socio-economic determinants, has both direct and indirect substantial implications for: underachievement in school, lower self-esteem and risk of developing cardiovascular disease or type 2 diabetes into adulthood.[4]

Health inequities contribute to the risk of nutritionally related diseases across the life course, such as diabetes, coronary heart disease and allergies.[5–11] Prevalence of childhood obesity is twice as high among those living in the most deprived, compared with the least deprived areas of the UK[12] and a similar trend is observed elsewhere in Europe.[13] Furthermore, acculturating to a host country has been shown to have significant impact on obesogenic behaviours. Migrants from low-income and middle-income countries moving to high-income countries tend to be healthier than their local counterparts in the host country when they newly arrived, known as 'healthy migrant effect'. However, immigrants post migration have been found to have poor health and more susceptible to be obese or overweight compared with the host population.[14] Some of the highest burdens of inequity and obesity are measured among ethnic minority populations, however, a tendency to aggregate data from South Asian (SA) populations limits a full understanding of trends that may result from different sociocultural norms and practices. Previous studies specifically examined nutritional outcomes in the British-Bangladeshi population reporting an increased risk of vitamin D deficiency[15] and iron deficiency,[16] which are essential nutrients for brain development in children; the highest proportion of obesity (31%) of any ethnic group among 6-year-old boys[17]; and poorer oral health in preschool children.[18]

The first 1000 days of life has been considered a window of opportunity.[19] Interventions delivered in the early years, and more specifically during the first 1000 days, to optimise nutrition and feeding practices could provide a holistic and potentially cost-effective approach to the prevention of nutritionally related diseases.[18] Extensive literature exists on barriers to exclusive breast feeding, yet comparably little attention has been offered to complementary feeding and care practices. The 6–23 months age period is known as the 'weaning' or 'complementary feeding' age, when solid and semisolid foods are nutritionally required to support breastfeeding. In 2018, the Scientific Advisory Committee on Nutrition (SACN) published the much-anticipated 'Feeding in the First Year of Life' report. Although many recommendations remained unchanged, it recognised the importance of feeding practices and caregiver behaviours such as familiarising infants to a range of flavours and textures.[20]

One forthcoming challenge is how to translate SACN guidelines into effective infant feeding programmes. Evidence demonstrates that culturally sensitive infant and young child feeding (IYCF) programmes need to be tailored to the target population and inclusive of the extended family and friends that typically offer more regular and often traditional support.[21] Although the National Health Service (NHS) has long recognised the value of community-centred approaches, few studies explore cultural and social drivers of complementary feeding practices to inform models that use local resources and assets.[22 23] There is an opportunity to learn from low-resource approaches that have proven to be effective globally[24–26] and build local partnerships to develop tailored community-based approaches. The results in this paper represent the formative phase of a wider programme of work Nurture Early for Optimal Nutrition (NEON).[27] NEON 1 initially engaged with British-Bangladeshi communities to develop generalisable evidence and an adapted Participatory Learning and Action (PLA) group approach. This was subsequently tested for its acceptability in the community and to identify any refinements that may be required in the approach (delivery and content of the intervention). The approach will be followed by NEON 2 where it will be delivered to other SA communities with the aim, if successful to potentially other diverse minority ethnic populations in the future to ensure a more inclusive approach to health promotion interventions.

## METHODOLOGY
### Study design
The NEON 1 is a National Institute for Health Research funded study that aims to explore common complementary feeding and care practices and their social and cultural influences within the British-Bangladeshi population of Tower Hamlets. NEON 1 follows a sequential design for evidence generation, beginning with a systematic review series of the literature on infant feeding and care practices in Bangladeshi and other SA families living in: India, Pakistan, Bangladesh and high-income setting.[28–32] This paper presents the results of the subsequent qualitative formative study phase, with participants from 'community', 'key informant', 'health professional' and 'family' groups (across generations), as informed by the multiple levels of the socioecological framework.[33] Evidence from the literature and qualitative study informed an adapted PLA group approach[34] to optimise infant feeding and care practices among Bangladeshi infants aged 6–23 months and will be described elsewhere.

### A community-based participatory research partnership
Informed by a community-based and participatory approach,[35 36] community facilitators (CFs) were active partners in study design, data collection, analysis and interpretation, towards intervention development. CFs worked as partners in the study team and as a bridge to the community, helping to facilitate engagement, crossing language barriers and informing accessible and appropriate topic guides. CF roles were advertised through local community organisations and parent

networks. CF recruitment criteria were (1) adult over 18 years of age, (2) parent, (3) self-identified as being of Bangladeshi heritage and (4) living in the local area at the time of the study: Tower Hamlets, London. Two CFs (one male, one female) were selected via interview according to their awareness and motivation towards the study and topic. CFs received an orientation day, a refresher day and 2 days of full training on interview and facilitation skills, data confidentiality and informed consent.

## Patient and public involvement

The NEON study was initiated in response to a clinical need identified by the UK SA community and healthcare professionals. A coproduction prioritisation exercise conducted with said groups identified the need for further research into obesity and diet within the UK SA population.[37] Furthermore, the epidemiology of the British-Bangladeshi community in Tower Hamlets (TH) and importance of the first 1000 days of life in relation to child growth and development provided a strong rationale for research and subsequent intervention with this population. Finally, the TH Clinical Commissioning Group (CCG) specifically identified the complementary feeding period (from 6 to 23 months) as an area lacking support and interventions within the borough, with breast feeding and child health already addressed through a range of borough-wide services and interventions. Once the need was identified by both the community themselves and the local CCG, patient and public involvement (PPI) representatives were identified and engaged in every step of the study from protocol development, to study design and dissemination of the results.

In addition to working in partnership at every stage with the CFs, PPI representatives were invited to attend quarterly research meetings and provide cultural oversight and study accountability. PPI study representative roles were advertised through local community organisations and parent networks. PPI study representative recruitment and selection criteria were the same as the CFs. Two PPI study representatives (both female) were selected.

CFs and PPI representatives received high street vouchers to thank them for their involvement and travel costs. The value varied according to length of the activity and guidance from community-based research partnership experts and involvement.

Community and health professional dissemination workshops were conducted in June and September 2019, respectively, with study participants invited to attend. Key findings, recommendations for an adapted PLA group approach and next steps, were presented and discussed with workshop participants, allowing for verification of identified themes and identification of ways to move forward to address suboptimal infant feeding and care practices and contributing factors within the study population.

## Study population

The study is located in the London Borough of TH, an area with the largest British-Bangladeshi population in the UK and one of the most socioeconomically deprived of any London borough, scoring the highest rates of child poverty and unemployment.[38] Sylheti is the most common language spoken by British-Bangladeshi residents of TH but is not a written language, meaning health promotion materials are usually written in English or Bengali and therefore limiting access to important information.

## Participant recruitment

Multiple recruitment strategies were used to purposively sample the British-Bangladeshi population of TH. CFs shared study details with local schools, community and children's centres and NHS clinics within different districts in TH ranging from Whitechapel to the Isle of Dogs. To ensure access to families that might not otherwise access health services, CFs also actively recruited from public spaces and through informal networks and through word of mouth. Posters were developed in English and Bengali, and placed in public spaces, however, CFs were encouraged to recruit via word of mouth to overcome language and literacy barriers. Key Informants were identified during community focus group discussions (FGDs) and through snowball sampling. Health professionals were identified and contacted by the researchers. Potential participants were approached by CFs in person or via telephone and provided with information and expression of interest documents. CFs arranged a venue, time and date convenient to participants, utilising community centres, children's centres and other public spaces in TH. The researcher organised all financial costs related to room bookings, transport and snacks.

## Sampling and inclusion

Study participants were eligible if they self-identified as British-Bangladeshi and were resident to TH, with the exception of health professionals who were only required to practice in TH. Inclusion criteria further differed by participant type. Community members were not required to have children in the study age range (6–23 months). Community key informants were included on the criteria that they held a significant role in the community. Health professionals were eligible if they had experience of nutritionally related diseases among infants and young children in TH; professions represented in this study included health visiting, midwifery, speech and language therapy, dentistry and general practitioner (GP). Finally, participants of the family phase were eligible if they were: (1) pregnant or (2) caring for a child aged 6–23 months in the capacity of: mother, father, caregiver or grandparent. Pregnant women and mothers were stratified according to time spent living in the UK (<or > 3 years).

## Focus group discussions

FGDs were first conducted with community members to explore social norms towards infant feeding and identify

stakeholders influencing infant feeding and care practices in the target population. FGDs were later conducted with fathers and grandparents to gain a consensus of understanding and inform findings from semistructured interviews (SSI) conducted with mothers and pregnant women. As advised by CF/PPI, all FGDs were separated by gender and status, that is, 'fathers', 'grandmothers' and 'grandfathers' to ensure a comfortable environment for discussion. FGDs were cofacilitated by the researcher LB and a CF. The gender of the CF was selected to match that of the FGD participants. One person acted as facilitator while the other took observational notes and this varied according to the study phase and group. At first, the researcher (LB) facilitated the groups but as the study progressed, CFs were capable of running the FGDs and the researcher observed. The researcher led the informed consent in English and CFs translated into Bengali or Sylheti. FGDs were conducted in community centres around TH and lasted approximately 90–120 mins. Audio recordings were taken for transcription and translation and all recorded interviews were then destroyed.

### Semistructured interviews

SSIs were conducted with key informants, mothers, pregnant women and healthcare professionals because this approach allowed a more in-depth exploration of infant feeding and care practices and influencers among target beneficiaries of this study. We sought to explore more in depth cultural and social norms towards infant feeding. These norms may change after migration to a new country[39] and thus we stratified recruitment of mothers and pregnant women by migration status: based in the UK<3 years or >3 years. The CF typically led key informant, mother and pregnant women SSIs in English, Bengali, Sylheti or a mixture of languages in community spaces or in the person's own home. Health professional interviews were conducted in English and in an NHS space or via phone, by the researcher (LB). SSIs were conducted individually except for one mother interview, where her mother and sister also attended. SSIs lasted approximately 45–90 min and were audio recorded.

### Topic guides

Topic guides followed a consistent structure but were tailored to the participant to encourage a progressive depth of information discussed. The literature[28–32] informed community and key informant and health professional interview tool guides. 'Family' topic guides were modified based on findings from these prior phases and tailored to family members (e.g. pregnant woman or mother). Participants were asked their thoughts on what foods they consider as healthy, and what a healthy child looks like and why, except for health professionals, who discussed influencing factors and health consequences encountered in their profession. Prompts were prepared and used for social, cultural and environmental influences on infant feeding. Guides were developed in English and translated into Bengali. Translations were checked and

edited by CFs to ensure reliability and consistency. CFs were prepared to speak in Sylheti, Bengali or English, as per the needs of the participants.

### Film

Stories and anecdotes collected during the early phases of data collection were used to produce a short film, introducing the topic of the study and stories to parents and caregivers that might be otherwise difficult to share to, for example, the narrative from one mother of filling a bottle of milk with crushed biscuits to 'top up' her infant feeding. The film was used to help with ice breaking during individual interviews with parents and pregnant women. The use of film has been recognised as a useful tool to support participants in feeling less threatened while yet familiarising them with the issue to be discussed. The films were collected in partnership with the community and only those films were used where consent had been provided for use in such settings.

### Ethical considerations

During FGDs and interviews, the researcher led the written informed consent in English and CFs translated into Bengali or Sylheti. The information sheets were translated into audio recordings in Bengali and Sylheti dialects. Translation and recording of study materials were carried out by CFs.

### Qualitative data analysis

Transcripts were checked by the researcher for fidelity of the topic guide and detail of prompts as data collection progressed. The steps of a framework analysis process were followed, as outlined by Clarke and Braun[40] and included: transcription, familiarisation with the interview, coding, developing a working analytical framework, applying the analytical framework, charting data into the framework matrix and interpreting the data. Transcriptions were completed by the CFs and outsourced to a professional company, parallel to data collection. CF and the researcher (LB) discussed and agreed when data saturation had been achieved. A combined inductive and deductive approach was taken to allow for socially located themes in the assumption that unspecified traditional and cultural beliefs are likely to influence feeding practices in this population. Two researchers independently familiarised themselves with the data and discussed preliminary themes and codes, to decide on the final themes and subthemes. Researchers applied open codes to the transcripts using NVivo V.11[41] and sought CF input into thematic analysis, to revise themes where applicable.

### RESULTS

A total of 141 participants were recruited to 12 focus groups and 45 SSIs (table 1).

Two overarching themes were identified during interviews and group discussions: (1) modifiable infant feeding and care practices that participants suggested

**Table 1** Characteristics of study participants

| Participant | FGD/SSI | Sex | N (Total) | Language |
| --- | --- | --- | --- | --- |
| Community members | FGD (n=6) | Female | 7 | Sylheti |
| | | Male | 8 | English with some Sylheti |
| | | Female | 5 | English with some Sylheti |
| | | Female | 10 | Sylheti |
| | | Male | 7 | English with some Sylheti |
| | | Female | 10 | English with some Sylheti |
| Key Informants | SSI | Mixed | 6 | Mixed |
| Health professionals | SSI | Mixed | 9 | English |
| Mothers | SSI | Female | 21 | Mixed |
| Pregnant women | SSI | Female | 9 | Mixed |
| Fathers | FGD (n=2) | Male | 6 | Mixed |
| | | Male | 4 | Mixed |
| Grandmothers | FGD (n=2) | Female | 14 | Mixed |
| | | Female | 14 | Mixed |
| Grandfathers | FGD (n=2) | Male | 6 | Mixed |
| | | Male | 5 | Mixed |

FGD, focus group discussion; SSI, semistructured interview.

could be targeted in order to optimise infant feeding and nutrition; and (2) socioecological factors believed by participants to influence these modifiable feeding practices. Online supplementary table illustrates the themes identified and example quotes from focus groups and interviews with participants.

Online supplementary table: Themes identified and example quotes from focus groups and interviews with participants 'online supplementary table'.

### Theme 1: modifiable infant feeding and care practices

Figure 1 depicts the four subthemes identified, namely: 'early and late introduction of semi and solid foods'; 'overfeeding'; 'feeding to fill the belly' and 'prolonged parent-led feeding practices'. Within some of these subthemes, there were further connecting subthemes as depicted in figure 1. For example, overfeeding included four connecting subthemes: forced feeding, unregulated portioning, top-up feeding and distraction feeding.

### Early and late introduction of semi and solid foods

Parents were highly aware of NHS recommendations to introduce semisolid and solid foods at 6 months. We identified differing attitudes towards this recommendation, favouring early (3–5 months), timely (6 months) and late (7–8 months) in different households, and a sentiment that specific advice regarding timing is 'always changing'. Early introduction was favoured among many participants who considered that early introduction would

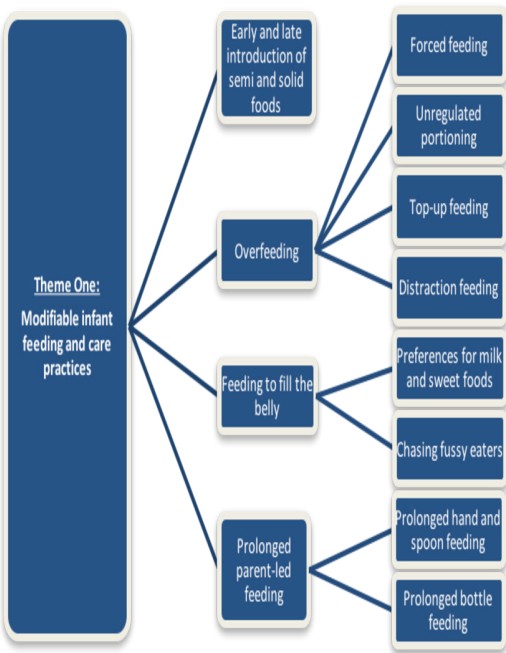

**Figure 1** Connecting subthemes related to theme 1: modifiable infant feeding and care practices.

improve the nutrition of non-breastfed babies or help their infants to grow and develop faster. Food advertising was given as a reason for the early introduction of solids from 4 months of age in a few households. Conversely, only a few parents introduced solid and semisolid foods later than 6 months and did so because they perceived milk as nutritionally sufficient. Late weaning was not as typical as early weaning.

> As a Bengali we tend to put our children onto solid food as quickly as possible.
>
> -Mother

Reasons cited by caregivers to introduce solid foods (irrespective of timing of introduction) included: the personality of the child, showing interest in food, health professional advice, when infant starts putting things in their mouth and when the infant is sitting up or crying for food.

### Overfeeding

Awareness on how to regulate feeding portions was mixed, with some parents stating that infants are able to self-regulate and others stating they 'don't have the understanding of fullness'. Four approaches to feeding are described below under the subtheme of overfeeding.

### Forced feeding

Key informants and health professionals defined 'forced feeding' as consistently encouraging a child to eat when not hungry. Parents did not report force-feeding themselves, but some said they had seen examples in their family or community. For example, holding a child in arms and feeding via parent-led practices, when the child is not thought to be hungry. As a nuance to the theme,

several mothers and fathers reported parents need to 'work hard and not force the child' because it can be an unpleasant or stressful experience and risks making children afraid of food. Those parents said they stopped feeding when the infant/child pushes the spoon away, refuses food, says 'no' or closes their mouth.

### Unregulated portioning

Family and community participants agreed that portion size is not customarily measured and often based on intuition. Health professionals reported that portion size can reflect adult portioning and that parental awareness, and a high frequency of feeds are barriers to age appropriate portioning. Parents felt that children were not eating enough volume of food when they took 'a half portion' or '2–3 spoons' per feed. One health professional addressed this concern in their own clinical practice (unrelated to this study) by filming in several households and measuring portion size and comparing perceived portion size with measured portion size. They found that measured portion size differed with perceived portion size frequently underestimating actual portion size (personal communication). Exceptions were found, however, among some parents, usually those that had received support from health services or sought information online.

### Top-up feeding

Regular 'top-up' feeding was defined as the practice of offering frequent and additional meals to an infant's feeding routine. Meal timings varied between households with a tendency to eat late into the evening (20:00 hours or later), providing time for extra feeds before infant/child bedtime. Some parents in this study said they gave their children milk and sometimes solid (jar) food immediately before the children go to bed. Occasionally, caregivers reported adding sugar or biscuits to milk bottles to encourage calorie intake between solid feeds.

> The other thing is obviously a lot of the people say right in our community, which is true, people eat up to the brim of their neck and even sometimes you will see people eating just after 15 min before they go to bed which is not good because you have not digested your food yet, it is still in your chest.
>
> - Community member

### Distraction feeding

Distraction feeding described the caregiver's practice of distracting the infant/child with videos on the phone or television (TV) while feeding directly, for example, via hand or spoon. Parents reported mixed attitudes towards the importance of play during meal times: some felt they benefited by avoiding messy eating and from the convenience of using distraction techniques, while others viewed this as a missed opportunity for interaction between caregiver and child. The TV was perceived as a helpful tool for grandparents with limited mobility to spend time with a child and supported mothers to undertake other household chores. A smaller number of caregivers described other distraction techniques including singing and storytelling while at the same time allowing or encouraging the infant to make a mess at meal times.

> [...] you know some babies they don't like to eat, almost you have to make different circumstances to make them happy to eat, like some entertainment thing like rhymes or something. Or lots of toys to put in front of her high chair and then [...]
>
> - Mother

### Feeding to fill the belly

Community and family members described how caregivers often feel a need to 'fill the belly', to ensure good growth and development. Some health professionals elaborated on this theme from their experience of parents that place an emphasis on feeding 'enough'. At most extreme, this can be taken as a literal demonstration in which community members said that caregivers might look for a physical protrusion of the belly. Awareness of fussy eaters appeared to motivate some parents to practice a range of approaches during mealtimes.

> There have been times, many, many times we've been telling parents how their baby's stomach is probably this small, like the size of their fist, that's how big your stomach is. So, if you are trying to feed them that much rice on a plate, that is more than they should be taking in. Parents tend to think oh no she didn't eat properly, she hasn't had 5 bites and they want to give more bites in her mouth or encourage to eat more. But really that is all the food she can take.
>
> - Community member

### Preferences for milk and sweet foods

A desire to fill the belly influenced the types of food and drink that parents preferred to give their infants, introducing formula milk, high energy (prioritising rice-based foods) and sugary snacks (e.g. yoghurt, banana, crisps or other snacks).

> I never know, my husband always says that's enough, don't give no more. He likes you know Asians make this thing called Kheer, it is milk with rice and sugar and it is really sweet and it is really nice. He absolutely loves it. Any time of day you feed him that he'll eat it.
>
> - Mother

Community members, informants and some parents recognised sugar as a customary part of the British-Bangladeshi diet. Community informants described a tradition to provide dates as the first food. Families described how sweets and treats are customarily shared by household visitors, and health professionals addressed the consequence that treats are frequently visible or on display in the household. Some mothers reported treats, including custard tarts and biscuits or fizzy drinks, were

commonly shared with young children. On the contrary negative attitudes to fizzy drinks did prevail among some study participants.

Mothers reported regularly preparing fruit juices, either as fresh, using a blender, or purchased, for example, mango juice from concentrate. Caregivers considered homemade smoothies and fruit juices as a healthy breakfast, lunch or snack but health professionals connected these preparations to a high daily sugar intake. Health workers described a tendency to wean onto the bottle using fruit juices and fruit squashes. More informed participants were aware of the risk to oral health posed by sugary diets, but still felt unable to break their dependence on sugar.

> I just don't want him to have too much sweet, because obviously all the kids in my family had all teeth taken out. I don't want him to go through that, he's only got five. I don't want him to lose all his beautiful teeth. I try and keep him away from sweets. I say that but I want to keep him away from sweet stuff
>
> - Mother

### Chasing fussy eaters

The phenomena of fussy eaters were described by participants at all levels as: children running away from or denying food, avoiding vegetables, avoiding solids, sticking to one kind of food or only taking three spoons against an expectation of four or five. Commonly described scenarios were of parents following children around, trying to feed them with their hands. Self-proclaimed parents of fussy eaters described their anxiety and concern for the dietary diversity of their children.

> You put him in his highchair and give him a few spoons and that's it. No, turns his face. And I have to like literally beg him, please one more, one more, he won't eat. And he's quite a healthy child. You'd think he eats quite a bit, he doesn't. He drinks 8ozs of milk and for him that's enough, he doesn't want no more.
>
> - Mother

Parents shared a range of concerns around the potential future health of their child and some spoke to the risk of vitamin D and iron deficiencies when asked if they provide their child with any supplementation. A concern around the risk of vitamin D deficiency stimulated one mother to switch to formula milk. Respondents from the community, health professional and family groups pointed to a cycle of fussy eating and the consumption of milk rather than solid foods. Health professionals observed that meal times can last around an hour, or could be replaced by sugary snacks and treats, perpetuating a cycle of milk and sugar but no meal. They recommended early exposure to a variety of textures and repeated attempts to introduce new foods (e.g. they described routinely advising families that it can take up to 14 attempts to accept a new food item) and advised against distraction feeding as a risk towards overfeeding.

> […] especially in the Bangladeshi community they have a Vitamin D deficiency so mum milk has everything but if the mum doesn't have enough vitamin D, the baby is not going to get enough vitamin D, so doctors are saying that formula milk has vitamin D, so mixed feeding is good. I talk to the doctor and my health visitor, they all listen to me. The formula milk has vitamin D specially made for the deficiency and so yes, I need to give him formula milk.
>
> - Mother

### Prolonged parent-led feeding practices

Infant feeding and care practices were predominantly parent-led (hand feeding, spoon-feeding, bottle-feeding, syringe) with few parents practising infant-led (ie, with finger foods) feeding.

### Prolonged hand and spoon feeding

Hand feeding was practised by parents as a symbol of love from parent to child and because they felt it could improve the taste of food while also allowing parents to remove small bones and to avoid messy feeding.

> I think because my Mum's always hand feed us, even when we were quite old she would hand feed us, […] I'm sure that Mum's here have hand fed their kids […] my son is five years old and I will hand feed him because rice and curry is mostly hand fed. Also, it's the love as well, it's the love aspect. It's like you're showing them you love your child, so then you feed them more. More than the love aspect is the easiness [… ] instead of them making a mess you just feed them […]
>
> - Community member

Health professionals and some parents raised concerns for extensive hand and spoon feeding over a prolonged period because it is considered a missed opportunity to encourage self-feeding practices and cognitive-motor skill development. The social impact was described by health professionals in a commonly described scenario, where some British-Bangladeshi children were described as being unable to use knives and forks when they reach primary school.

> Culturally we've always eaten rice by hand, okay. That's something I would never let go of. In all honesty, rice and curry is so tasty with hands, we don't enjoy it with fork and knife.
>
> - Mother

Spoon feeding was preferred for semisolids, such as porridge or khichdi (a rice and lentil preparation) or during teething, while the syringe was less commonly used and mostly for water when the child was considered to have digestion problems. Parents using beakers were also following advice to introduce at 6 months and often use it for water but also used it for juices such as Ribena. Infants were fed in a range of positions, often depending

on the method of feeding and location. Inside the home infants may sit in a high chair at the table or separately, on the lap, floor, sofa or caregiver's arms. A common scenario was described of 'chasing the baby around the sofa', although this was seen as less common.

> I saw most of the mothers running behind children with a spoon
>
> - Mother

### Prolonged bottle feeding

Bottle feeding was frequently reported among British-Bangladeshi families in this study. Parents reported that the bottle was introduced as early as a few weeks or months in households that followed a mixed (breast and bottle) feeding approach. Occasionally, parents were motivated by a belief that formula milk is scientifically prepared and thus more beneficial for the child. Many mothers introduced the bottle after birth to facilitate feeding outside of the house, during social situations or when visitors come to their home. Typically, the bottle was viewed as a stepping-stone towards the introduction of solid foods. Friends and family often encouraged bottle feeding to ensure the 'belly is full', as it was felt to allow the possibility of more frequent feeds, such as an extra feed before bed to help the infant/child sleep and allows family members to help free up the mother's time.

> My boy is used to the bottle from his father, maybe at just one or two months he did breastfeeding, after that he didn't breast feed now he is sleeping my wife tried the breastfeeding at night time. Sometimes he drinks but sometimes not, but he used to have the bottle and it's easier for us as well to give him the bottle.
>
> - Father

There were various accounts of how community members who had seen the bottle being used in a suboptimal way, for example, the contents being modified to include sugary additives, such as biscuits, or to give juice such as mango to infants less than 6 months of age. One community member recalled a friend had left a bottle of juice in the mouth of an infant overnight, something that she was concerned about for the impact that it could have on tooth decay. Another community member described 'modified bottle' practice included cutting the teat of the bottle to feed semisolid foods.

One health professional recommended a complete transition away from the bottle by 1 year of age, but a form of prolonged bottle feeding was described by health professionals among children up to the age 2 or 3 years old in many of the households. A speech and language professional identified prolonged sucking on a bottle teat as a risk factor for impeded development in the roof of the mouth of some British-Bangladeshi infants in their care. Health professionals also expressed concern that prolonged dependence on the bottle may suppress the

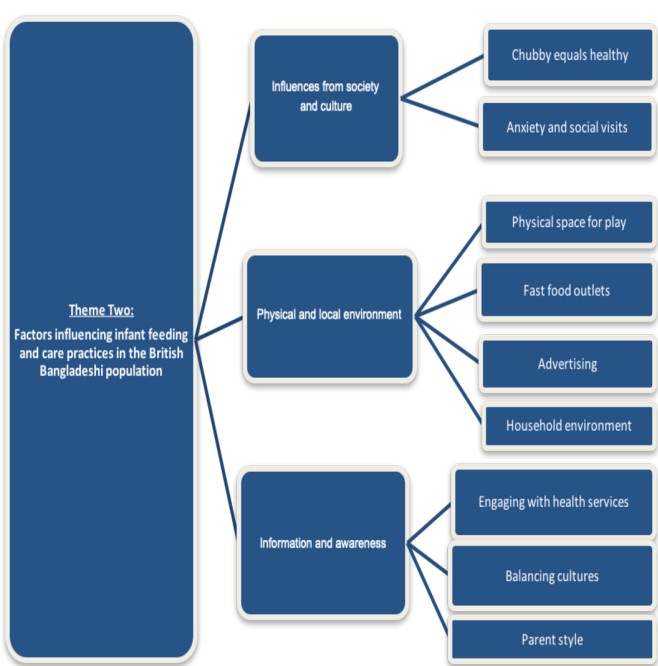

**Figure 2** Connecting subthemes informing theme 2: factors influencing infant feeding and care practices in the British-Bangladeshi population.

appetite and that it is linked with an additional issue of dietary diversity of children consuming excessive milk.

> A third of our children are on the SEN [special educational needs] Register which is speech and language. They are still probably having bottles, they are probably still on dummies and they probably aren't eating solid foods properly and they are being (hand) fed as well. That's the other thing in our community as well that children up until the age of about 7 or 8, even older, are physically fed rice, you know, hand fed. When children come into nursery […] some of them don't even have a clue about how to use a knife and fork because they have been fed. When the food is fed […] rice is mushed with their hand […] it doesn't give them that whole experience of the chewing.
>
> - Health professional

### Theme 2: factors influencing infant feeding and care practices in the British-Bangladeshi population

Figure 2 illustrates the wider socioecological factors identified by participants that they thought impacted on infant nutrition that fell into three subthemes as depicted in figure 2: (1) society and culture, (2) physical environment and (3) information and awareness.

### Influences from society and culture
#### Chubby equals healthy

Community members, key informants and parents presented an archetypal societal image of a 'chubby', healthy child. Community members rooted this finding in a belief that extra weight safeguards growth and development during childhood—and this is something

that was echoed in the discussions with parents. Parents placed extensive emphasis on good growth and development, which was experienced by health professionals as a concern for appropriate growth along WHO growth centiles. Health visitors described having guided some mothers to develop a routine to support their meal planning and build resilience against outside community perceptions of a healthy weight. Some parents have learnt about the importance of responsive feeding by attending local courses. Community members explained that past generations had experienced a scarcity of resources when raising children in rural Bangladesh and this legacy continues to influence modern feeding practices, although resources are greater and physical activity is lesser in urban East London.

> The baby who has good health with no disease, is chubby. With good health they look nice.
>
> - Pregnant woman

### Anxiety and social visits

Mothers reported some anxiety towards social occasions and particularly if they anticipated comments from extended family members or friends, suggesting their child was too skinny, as a critique of not feeding enough, or too chubby, when perceiving jealousy (see quote). Mothers reported occasionally finding it difficult to challenge social practices, such as offering treats in other people's households, even when they avoided such norms in their own homes.

> If we try to talk to her about [overfeeding her child] she gets offended. In Bengali culture, we have what we call the 'evil eye' […] so, she thinks we are giving her children the 'evil eye' because they eat well, they eat a lot. We still talk to her about it [overfeeding] though, we don't stop.
>
> - Mother

## Physical and local environment

### Physical space for play

Community members, especially fathers and grandfathers, felt that limited household space left little opportunity for play in some households. The limited physical activity of most family members was often considered to begin in childhood.

> If you want to be healthy you've got to start from the beginning. Here what do you do? People are living in a box, just in the house or flat, got very limited space for a young child, 9 months or 1 year old child, it's time for them to crawl around. Is there any space to do that? No. Their bone structure, everything, is not getting stronger because they are not exercising, they are not moving around that much. What they do, say put them in a pushchair, in front of the TV and that's it.
>
> - Community member

Elevated levels of outdoor air pollution and concerns for safety in public spaces on rare occasions were added barriers to exercise. Furthermore, exercise was not considered part of the daily routine, this being enhanced by a recognition of the shift in lifestyles after migration from very physical in rural Bangladesh to more sedentary lifestyle in urban UK. However, some parents did encourage swimming, football or taking walks in the park, facilitated by access to facilities and female-friendly sessions to try and encourage some engagement with exercise.

### Fast food outlets

Community members widely regarded the abundance of fast food outlets in the TH environment as a well-established influence of takeaway meals such as chicken and chips. Some parents reported having a weekly takeaway as a treat. Families also tended to share foods with infants/children and allow them a small taste of these meals which were often high in salt and fat.

> […] my grandchild is nearly 6 years old and we did try a little bit of chips to taste it.
>
> - Grandfather

### Advertising

Parents found the advertising of semisolid and solid convenience foods from the age of 4 months as confusing and indicative that it is an appropriate age to try solid foods. The availability of formula marketed for 'hungrier babies' supported some parent perceptions that their baby needed more food or more energy dense foods than other babies.

> […] I think anyway if you go to the supermarket and you see something that says four months then you should be doing that because the packaging, if the baby food companies are telling you that you should be feeding your baby at four months why wouldn't you, why would you wait until they were six months, because even though it is not that much of a difference but when you have had a baby they are growing so quickly you think that two months is a big deal and that you should be giving that extra food, but definitely I do not think many families that I have seen will wait until six months.
>
> - Community member

### Household environment

The visibility of fruit and vegetables in the household was seen as an easy barrier to overcome in many households. Health professionals and key informants within the community observed that fresh fruit and vegetables are not usually on display in households. Conversely, sweets may be on display in the home during social situations such as family gatherings. Health professionals advised helping to introduce different food types and textures by allowing for the visual and touch experience, beyond just taste.

[…] Especially around holiday time everyone goes to everyone's houses and things like that. A lot of the dishes would be made, obviously when they are doing these kinds of things you are not looking at what you are cooking, what kind of ingredient is going in, what level of salt or sugar has gone in. That is when children do eat quite a lot more than what they are supposed to eat when you have got guests coming into your house they are bringing like sweets, crisps and things like that for your children.

- Key informant

### Information and awareness
#### Engaging with health services
Mothers and Fathers differed in health seeking preferences; fathers stated a preference to speak to a GP while mothers described preferring to seek information from sources including Health Visitors, Children's centres, online and community groups. Health professionals and informed community members reported a range of demonstrations and activities for new parents offered at local health and community centres and fundamentally supported with a crèche facility. However, parents and carers widely perceive that advice is constantly changing or conflicting or simply not available. Pregnant women felt antenatal classes would help to explain what to do but not how to feed after birth and fathers sometimes felt ignored by GP's. In general, parents reported wanting more support on topics ranging from: how to introduce solid foods, when to introduce semi/solid foods, when to stop giving milk, what foods to give, in what ratio and how much.

I am always confused, I am always ringing up the doctors and asking them questions all the time, especially about the milk thing. They said it depends on your child, every child is different. I was a bit like he's still little, doesn't he need the nutrients in the powder to grow?

- Mother

First time parents especially wished to have more information and support. Health professionals reported that complementary feeding and care needs to become a routine aspect of post-natal care services and that many would benefit from a more consistent 'hand-holding' approach that is currently lacking.

#### Balancing cultures
Mothers often tried to interpret information from different cultures and generations in their infant feeding practices. Grandparents and in-laws were viewed as either supportive influences or pressurising depending on the household. Parents reported adverse feeding practices among extended family members, including: feeding extra meals, feeding before bed and placing children in front of the TV.

By starving, I actually used that term, starving may mean they think the child has not eaten […] like for example if they have not eaten in two or three hours […] I will give you a perfect example, I went to the wedding over the weekend and my mother in law was on my case to feed my son […] I just said when he is hungry he can communicate with me, he will come up to me and say mum I want to eat. I tried three times, after three times I am not going to go round walking after him. My mother in law you know how she is she is so protective of her grandchildren […] she took him and she fed him, so can you see the difference?

- Mother

Grandparents reported having usually experienced very different advice and support while they were parenting their own children, to parents living in the UK and this appeared to create a generational gap in knowledge, lived experience, awareness and practice that was recognised by both generations. Some grandparents reported feeling like their children did not listen to their advice while some mothers and carers reported their mothers or mothers in law provided conflicting advice to that of health professionals. However, these opinions were not universal.

I remember before I got married before I had my kids they used to do what my mum did which my mum did twenty three years ago, twenty four years ago, at that time it was just egg custard and stuff so they did not know much about vegetables, they did now know much about a lot of stuff. Even when we were growing up probably my mum gave like at six months they started introducing what they ate but before that they gave more like egg custards

- Mother

Some mothers felt they needed to be 'very resilient' to continue breastfeeding, while others felt that grandmothers or mothers in law were supportive of exclusive breast feeding. Commonly, however, many mothers were proactive in seeking information from different sources, and some mothers felt they were able to resist unhelpful advice by 'being a strong person' and 'doing my own research and asking advice', or by having a close network of experienced mums.

[If] they are fussy eater, it is just because they were fed on only milk because of my daughter's in laws, they are afraid that house will be a mess. They do not know the foods, you have to try foods to know.

- Grandmother

#### Parent style
Mothers described experiencing anxiety when they felt they did not have enough or the right information and when they felt they did not know how to feed their child, a commonly expressed concern in mothers of 'fussy eaters'.

 Lakhanpaul M, *et al. BMJ Open* 2020;**10**:e035347. doi:10.1136/bmjopen-2019-035347

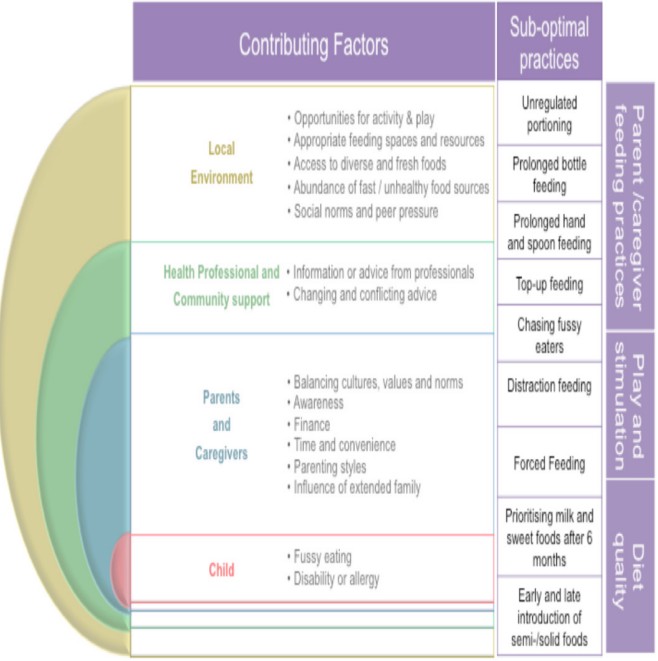

**Figure 3** A framework for priority infant feeding and care practices in British-Bangladeshi communities and the contributing factors that need to be addressed under a socioecological approach.

Yes, I am worried about him because he's not eating. I try and give him, if he eats anything like the other day I gave him a choc chip roll when I shouldn't have, like it's not good for him to have and he ate it and I was so happy. He never says no to chocolate though, that's a bad thing. Milky bar, his grandparents, always milky bars, he loves it.

- Mother

A more relaxed or confident approach was found among mothers that considered themselves as proactive in seeking information.

I notice in our community we have a real thing about feeding them and making sure the fact they are having a full […] do you see what I mean but I just tend to find that be a bit relaxed, see if they take to it, if they eat or whatever. If they don't eat, come back to it, come back to the same meal, come back to that thing. It might be you try something else, they might like it but don't become stressed by it.

- Mother

These mothers recognised the various sources of information and invested time in engaging with different groups, either online or in the community depending on their time, locality and interest. Some mothers considered themselves as a role model to their children and tried to adopt healthy practices to that effect or developed a daily routine with the support of health visitors.

## DISCUSSION

Two overarching drivers of infant feeding and care practices were identified in this qualitative study focusing on the British-Bangladeshi population of TH, namely: potentially modifiable parent infant feeding and care practices and wider socioecological factors. An assortment of potentially modifiable feeding and care practices were identified that may act to restrict dietary diversity, override infant satiety, facilitate a reliance on sugary or sweetened foods and drink, and encourage overfeeding. Common parent-led feeding practices make it difficult for many parents to pick up on satiety reflexes in the infant/child. Parents often feed when they perceive the child as hungry or to show affection in common practices such as: holding a baby in arms; distracting the child with a phone/TV; and prolonged parent-led feeding (via spoon, bottle or hand); however, such practices can override childhood satiety cues and impact appetite, oral hygiene and speech and language development in childhood and across the life course.

Figure 3 provides a thematic representation of factors that contribute to suboptimal infant feeding and care practices spanning the socioecological environment. Of note, the local food environment is a complex system known to shape individual diet.[42] Participants of this study described social, built and natural influencing factors of infant feeding (figure 3). Societal perceptions of 'chubby' as healthy was an especially prominent theme and discussed in tandem with anxiety among those that felt they were not following normative approaches to 'filling the belly'. Participants also emphasised the benefits of access to children centres and health visitors in TH for accessible support and problem solving. Challenges extended to the local environment, where participants identified a lack of opportunities for play and physical activity, appropriate feeding spaces, an abundance of fast food outlets and misleading advertisements of infant foods.

Acculturation represents an important construct that explains the determinants of health inequalities in ethnic minority groups. In this study, acculturation was found to have an effect on the nutritional feeding practices in the British-Bangladeshi community. This can be seen in the generation gap between mothers and grandparents: for example, the influence from the extended family on the way the mother feed her infant and the fact that past generations had experienced a scarcity of resources when raising children in rural Bangladesh, which continued to influence modern feeding practices, although resources are greater and physical activity is less in urban East London. This generation gap is part of the acculturation of the migrant with the host country where second generation were found to be more susceptible to follow the NHS and health advice. With that in mind, culturally sensitive health information could help to reduce the negative effect of acculturation and health disparities on infant nutrition and complimentary feeding practices.

## Consistency

Our findings are consistent with evidence of an association between ethnicity, infant feeding and care practices and body mass index among the British and SA populations in Bradford.[43] We add to emerging evidence from TH that identifies a myriad of factors, including family and friends as influencers of infant feeding and care practices in TH[44] and a systematic review that recognises the value of heritage, tradition and culture on infant feeding and care practices within SA families living in high-income settings.[31]

## Strengths and limitations

Culturally acceptable interventions to promote optimal infant feeding and care among minority populations have been relatively under-researched to date. Existing literature does not adequately capture the cultural and social determinants of infant feeding and care. A strength of this study is the development of culturally sensitive interview tools and recruitment strategies, based on evidence from the literature and guided by our PPI and CFs. This strong partnership with members of the British-Bangladeshi population of TH enabled further reach and diversity of representation in this study. A limitation of this qualitative study is that a quantitative measure of dietary intake was out of scope. However, cohort studies explore the association between infant dietary intake and ethnicity.[45] Another limitation is that all data are self-reported and have the potential for respondent bias. We believe we overcame this limitation through triangulation of respondent data, a phased approach that encouraged further probing into emerging topics between stakeholder groups and by collecting data from a relatively large sample.

## Implications for clinicians and policy-makers

These qualitative data describe a layer of influences and add strength to the debate around regulation and marketing of follow on infant formula and foods and availability of fast food outlets. In 2016, the UK launched 'Childhood obesity: a plan for action (2016)', in which the Department of Health set the target to reduce England's rate of childhood obesity within the next 10 years. The action plan focuses on partnership with the private sector but has been criticised for an inadequate stance on regulatory measures.[46] The sugar tax was a celebrated feature of the plan, but comparatively little political action has been taken towards early childhood nutrition or enforcing advertising regulations. Updated SACN guidelines emphasise educating parents and caregivers on behavioural practices relating to infant feeding, including introducing new foods, textures and diversity, yet this will take time to reach households. As such, the current model of early feeding support can be perceived as absent, untimely or confusing for caregivers and is not tailored and does not take account for cultural practices. Recommendations can, therefore, sometimes seen unrealistic or at odds with community practices and may have a negative impact on health seeing behaviour or engagement by minority ethnic groups.

More regular contact between pregnancy and 24 months, signposting to existing and local resources and additional support in the form of facilitator led PLA evidence informed women's groups could improve the current model of infant feeding support. The economic incentive alone for early nutritional intervention in the UK is profound; treating obesity-related disease is estimated to cost the NHS £6.1 billion pounds per year and the wider costs to society estimated at around three times this.[47] A growing interest in reverse innovation[48]—the concept of transferring a product or service developed in a resource poor setting to an industrialised setting[49]—has led to demonstrable cost savings for the NHS in areas from technology[50] to community mobilisation.[51] There remains a question of who and how to deliver effective community-based programmes under the NHS? Local leadership is essential; this study was fostered in part by a supportive local agenda, where the Health and Well-Being Board enshrine 'Children's Weight and Nutrition' as one of five top priorities. Similar boards provide a comparable opportunity for action and partnership with civil society and private sector if sustained by an ideology rooted in the social determinants of health.

## Generalisability

This study provides an in-depth understanding of cultural and social influencers of infant feeding and care within the largest national British-Bangladeshi population and supports observations from other SA populations in the UK.[43 45] As such, these findings may be generalisable or will be able to be used to inform similar discussion with British-Bangladesh residents across the UK as well as SA populations and other ethnic minority groups. A broad range of stakeholders, including health professionals, informed our understanding of the shared environmental challenges to infant feeding and care and need for improved IYCF that exist nationally. We recommend that this tailored approach to exploring IYCF practices be extended to other ethnic minority populations in the UK. Finally, while this study focuses on an ethnic subgroup, the dual burden of malnutrition is a global concern. Data on IYCF indicators are limited but the data that are available show concerns for complementary feeding practises across high, middle and low-income settings that are in line with our study findings. We suggest that while many themes may be generalisable, culturally tailored approaches will be required to explore IYCF in other settings. We seek to use this formative research to inform the adaptation of a participatory and community-based intervention in an NHS context, which could have far-reaching implications for programme design and vulnerable populations in high-income settings globally.

## Future research

This study identifies the need for more evidence to inform services that effectively optimise feeding and care in the

early years and target vulnerable populations in the UK. We recommend for more research to support the development of evidence-based interventions that optimise feeding practices across all ethnic majority and minority populations in the UK under a community-based participatory approach. We encourage culturally sensitive and community-based programming that facilitates families to discuss and challenge societal norms underpinning infant feeding practices, considerate of the whole family unit, including fathers and grandparents. More research is needed to inform an integrated participatory approach so that caregivers and community can feasibly be included in an NHS–community partnership to codevelop IYCF services. As the PLA group approach has been demonstrated to bring statistically significant improvements in maternal and infant survival through application with participatory women's groups in low-income and middle-income settings[26]; NEON, therefore, ultimately seeks to determine whether an adapted PLA group approach has the potential for early and effective intervention, using infant feeding as an exemplar.

## CONCLUSION

This body of work generates transferable lessons to inform similar studies for adapted or tailored programmes with other vulnerable populations. It also highlights the importance of wider determinants of health and behaviours and the role for policy and environment in addressing health outcomes in general, but in particular those high risk or more vulnerable populations. We recommend that effective, culturally sensitive support be provided to parents, caregivers and extended family members at different 'ages and stages' through evidence based and tailored infant feeding programmes. In addition, UK infant feeding environment requires better regulation of marketing of foods for infants and young children if it is to optimise nutrition in the early years.

**Author affiliations**
$^1$Population, Policy and Practice, University College London Institute of Child Health, London, UK
$^2$Whittington Health NHS Trust, London, UK
$^3$Department of Epidemiology and Public Health, University College London Institute of Epidemiology and Health Care, London, UK
$^4$Aceso Global Health Consultants Ltd, London, UK
$^5$Specialist Children's and Young People's Services, East London NHS Foundation Trust, London, Newham, UK

**Acknowledgments** The authors would like to thank the National Institute of Health Research, Collaboration for Leadership in Applied Health Research and Care North Thames for funding the NEON study. We would like to thank all NEON study team members for their contribution and guidance to research activities throughout the three phases of the NEON study. All study team members made considerable contributions to research activities and decision making throughout the duration of the study and supported identification of key study findings and recommendations stated in this report. We would like to thank the NEON study implementation partners, who supported and participated in research activities and enabled the research process to proceed effectively. NEON study implementation partners included Tower Hamlets Borough Local Authority, Barts Health NHS Trust, Newham Local Authority, Women and Children First, Women's Health and Family Services, and The Breastfeeding Network in Tower Hamlets. Thanks go to the NEON advisory board, made up of researchers, practitioners, study partners and patient public involvement members who attended NEON advisory meetings, offered guidance and supported decision making on study activities. We would like to thank the NEON study's community and Participatory Learning and Action group facilitators who performed integral roles in data collection and NEON intervention delivery activities. Finally, we would also like to thank the British-Bangladeshi population in Tower Hamlets for participating in and supporting the research.

**Contributors** ML was the lead investigator, conceived the idea, contributed to the tool development and drafting of the first draft of paper and edited further drafts. LB was lead research fellow, drafted interview tools, collected the data and wrote the first draft of the publication. OL-H was involved in interpretation of data, manuscript write up and dissemination of study findings. LM was involved in study conception, manuscript write up and dissemination of study findings to partners and the general public. DMR was involved in interpretation of the results and manuscript write up. SA was involved in interpretation of the results, manuscript write up and dissemination of study findings. MH was involved in study conception, interpretation of data, manuscript write up and dissemination of study findings.

**Funding** This study was funded by National Institute for Health Research.

**Competing interests** All authors have declared that no support from any organisations for the submitted work; no financial relationships with any organisations that might have an interest in the submitted work in the previous 3 years, no other relationships or activities that could appear to have influenced the submitted work.

**Patient consent for publication** Not required.

**Provenance and peer review** Not commissioned; externally peer reviewed.

**ORCID iD**
Monica Lakhanpaul http://orcid.org/0000-0001-5288-3325

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
