## [Reviewer comments · BMJ Open]

ARTICLE DETAILS

TITLE (PROVISIONAL)	Nurture Early for Optimal Nutrition (NEON) Programme: Qualitative study of drivers of infant feeding and care practices in a British-Bangladeshi population
AUTHORS	Lakhanpaul, Monica; Benton, Lorna; Lloyd-Houldey, Oliver; Manikam, Logan; Rosenthal, Diana; Allaham, Shereen; Heys, Michelle

VERSION 1 – REVIEW

REVIEWER	Annie Hardison-Moody NC State University United States
REVIEW RETURNED	26-Nov-2019

GENERAL COMMENTS	Introduction: In the introduction, the authors talk about how obesity leads to poor performance in schools and other outcomes; however, there are mediating factors that are impacting these rates (poverty, racism, etc.). I'd encourage the authors to not blame these outcomes on weight, but complexify this statement to acknowledge the broader socio-economic factors that shape these outcomes. In paragraph two, the authors talk about increased obesity for immigrant populations; in the US, immigrants arrive healthier and then over time their health declines. I don't know if it's the same in the UK (I'm guessing yes?), but it might be important to point this out as well as a key point in dietary acculturation theory. I think you all do a very nice job of setting up the need for this study (well done!), but wonder if you want to add any research on dietary acculturation, since I think it's relevant for the points you are making (esp. that we can learn from global research on interventions that work). Very glad you are doing this work on complementary feeding – my own qualitative research in the US demonstrated that so many mothers are confused and need better support around this. I think that you need a strong argument at the outset of the paper – you note that you identified some key themes, but you have a much stronger argument about the importance of culture and context, and some findings that you have related to that. Can you state that here? Methods I think that the last sentence of the study population paragraph could be cut or moved to conclusion. Really appreciate that you did focus groups with community members, not just mothers/caregivers. So much of this work is
--

focused on mothers, but as you know, they exist within a broader web of social, familial contexts.

Results

Page 15, line 10 – think a is missing before few

In reading the results, I think it might be helpful to break out some of the results by who shared them – ex, what did parents say about overfeeding vs. community members? It's not always clear if everyone thought these things, or just the parents. This will be relevant for intervention design, and would be helpful for the reader to understand. If they said similar things, I'd state that; if not, highlight differences. You've done this in some of the sections (ex: overfeeding, portions) but it's not as clear in others.

I wanted a lot more of the quotes and stories from participants, but you have a lot of data, so I'm wondering if you could include a table that follows your conceptual model categories, but with several representative quotes in each section. Then you could remove some of the quotes from the text and introduce a bit more nuance and detail to each of the themes. Some are quite short, but they are so important! For example, it would be good throughout to have a better sense of differences between groups (as noted above); whether themes were very prevalent or just seen among some of the families/mothers; giving a bit more detail in some sections (ex, page 27 top paragraph – explain a bit more differences in how mothers experienced grandparents).

Discussion

I think that your key findings are a little too sparse – there is a lot that is new/important in this research that you should highlight. For example, the findings about culture and religion are very important and definitely not well understood in the literature or practice (might help to review some religion and health literature, Journal of Religion and Health could be a place to start), or around acculturation and infant feeding. Rather than just a description, I'd like to see a clear argument here about the importance of this work and why it matters. I know that you get to that in the implications section, but the discussion as written now is a bit too repetitive of the results. Instead, use this section to highlight what is new and cutting edge from the literature. Ex, the consistency section needs to be stronger – make the point about how although your work aligns with current work, it's actually pushing the field forward as well.

Implications

See my notes above – this section doesn't really highlight your key argument or demonstrate how this work is advancing the field. As I'm thinking about it now, I wonder if a theoretical frame about dietary acculturation might help you to make a clear argument about these broader factors, and then allow you to make the point that clinicians/practitioners, etc. must take account for culture and these broader social and structural factors? That's just one idea, but I think throughout you need a stronger argument about the research findings than just noting two themes from the research. Tell us why this matters and why it's important!

	Conclusion – same as above. The article is making a strong contribution re: culture, social context and this needs to be stated more forcefully in the conclusion.
--	--

REVIEWER	MILKAH WANJOHI AFRICAN POPULATION AND HEALTH RESEARCH CENTRE
REVIEW RETURNED	04-Feb-2020

GENERAL COMMENTS	The paper is clear, well written and addresses an important issues around complementary feeding, especially on the influences and barriers to optimal CF in deprived neighbourhoods. I have included minor edits below; Page 5 Line 56 - check if 1001 or 1000 days, the WHO references indicate first 1000 days Page 6 Line 26 - define NHS as it papers here for the first time Page 7 Line 25 - add a reference for the social ecological model/framework page 11 , clarify if these were films from other study participants (in the previous study phases) and if so, indicate if these participants consented to have their films used for this phase.. Page 11 line 26, indicate if the FGD consent was written or verbal and whether group or individual Page 12, table 1 , the number of interviews adds up to 141 not 145 , please check if this is correct Page 17, the quote does not support the narrative on preference on sweet foods , please use a quote that is more reflective of the narrative.. page 31, the first paragraph in the discussion section is not necessary, since the information therein is already covered in the previous sections, and hence seems repetitive.. page 36, line 33 revise to 'made up of'
---

VERSION 1 – AUTHOR RESPONSE

Reviewers' Comments to Author:

Reviewer: 1

Reviewer Name: Annie Hardison-Moody

Institution and Country: NC State University, United States

Please state any competing interests or state 'None declared': None declared

Introduction:

In the introduction, the authors talk about how obesity leads to poor performance in schools and other outcomes; however, there are mediating factors that are impacting these rates (poverty, racism, etc.). I'd encourage the authors to not blame these outcomes on weight, but complexify this statement to acknowledge the broader socio-economic factors that shape these outcomes.

In paragraph two, the authors talk about increased obesity for immigrant populations; in the US, immigrants arrive healthier and then over time their health declines. I don't know if it's the same in the

UK (I'm guessing yes?), but it might be important to point this out as well as a key point in dietary acculturation theory.

I think you all do a very nice job of setting up the need for this study (well done!), but wonder if you want to add any research on dietary acculturation, since I think it's relevant for the points you are making (esp. that we can learn from global research on interventions that work). Very glad you are doing this work on complementary feeding – my own qualitative research in the US demonstrated that so many mothers are confused and need better support around this. I think that you need a strong argument at the outset of the paper – you note that you identified some key themes, but you have a much stronger argument about the importance of culture and context, and some findings that you have related to that. Can you state that here?

Reply: We appreciate the positive feedback from the reviewer, which have helped us greatly improve the quality of this manuscript. As suggested by the reviewer, we have made all the suggested corrections.

Methods

I think that the last sentence of the study population paragraph could be cut or moved to conclusion.

Really appreciate that you did focus groups with community members, not just mothers/caregivers. So much of this work is focused on mothers, but as you know, they exist within a broader web of social, familial contexts.

Reply: Thank you for your useful comments. We've incorporated all changes suggested above.

Results

Page 15, line 10 – think a is missing before few

In reading the results, I think it might be helpful to break out some of the results by who shared them – ex, what did parents say about overfeeding vs. community members? It's not always clear if everyone thought these things, or just the parents. This will be relevant for intervention design, and would be helpful for the reader to understand. If they said similar things, I'd state that; if not, highlight differences. You've done this in some of the sections (ex: overfeeding, portions) but it's not as clear in others.

I wanted a lot more of the quotes and stories from participants, but you have a lot of data, so I'm wondering if you could include a table that follows your conceptual model categories, but with several representative quotes in each section. Then you could remove some of the quotes from the text and introduce a bit more nuance and detail to each of the themes. Some are quite short, but they are so important! For example, it would be good throughout to have a better sense of differences between groups (as noted above); whether themes were very prevalent or just seen among some of the families/mothers; giving a bit more detail in some sections (ex, page 27 top paragraph – explain a bit more differences in how mothers experienced grandparents).

Reply: Thank you for your useful comments. We've incorporated all changes suggested above. We have also attached the quotes table as requested by the reviewer.

Discussion

I think that your key findings are a little too sparse – there is a lot that is new/important in this research that you should highlight. For example, the findings about culture and religion are very important and definitely not well understood in the literature or practice (might help to review some religion and health literature, Journal of Religion and Health could be a place to start), or around acculturation and infant feeding. Rather than just a description, I'd like to see a clear argument here about the importance of this work and why it matters. I know that you get to that in the implications section, but the discussion as written now is a bit too repetitive of the results. Instead, use this section to highlight what is new and cutting edge from the literature. Ex, the consistency section needs to be stronger – make the point about how although your work aligns with current work, it's actually pushing the field forward as well.

Reply: As suggested by the reviewer, we have made this correction; we have written about the acculturation and infant feeding.

Implications

See my notes above – this section doesn't really highlight your key argument or demonstrate how this work is advancing the field. As I'm thinking about it now, I wonder if a theoretical frame about dietary acculturation might help you to make a clear argument about these broader factors, and then allow you to make the point that clinicians/practitioners, etc. must take account for culture and these broader social and structural factors? That's just one idea, but I think throughout you need a stronger argument about the research findings than just noting two themes from the research. Tell us why this matters and why it's important!

Conclusion – same as above. The article is making a strong contribution re: culture, social context and this needs to be stated more forcefully in the conclusion.

Reply: Thank you for your useful comments. We've incorporated all changes suggested above.

Reviewer: 2

Reviewer Name: MILKAH WANJOHI

Institution: AFRICAN POPULATION AND HEALTH RESEARCH CENTRE

Please state any competing interests or state 'None declared': NONE

The paper is clear, well written and addresses an important issues around complementary feeding, especially on the influences and barriers to optimal CF in deprived neighbourhoods.

I have included minor edits below;

Page 5 Line 56 - check if 1001 or 1000 days, the WHO references indicate first 1000 days

Page 6 Line 26 - define NHS as it papers here for the first time

Page 7 Line 25 - add a reference for the social ecological model/framework

page 11 , clarify if these were films from other study participants (in the previous study phases) and if so, indicate if these participants consented to have their films used for this phase..

Page 11 line 26, indicate if the FGD consent was written or verbal and whether group or individual

Page 12, table 1 , the number of interviews adds up to 141 not 145 , please check if this is correct

Page 17, the quote does not support the narrative on preference on sweet foods , please use a quote that is more reflective of the narrative..

page 31, the first paragraph in the discussion section is not necessary, since the information therein is already covered in the previous sections, and hence seems repetitive.

page 36, line 33 revise to 'made up of'

Reply: Thank you for your useful comments. We've incorporated all changes suggested above.

VERSION 2 – REVIEW

REVIEWER	Annie Hardison-Moody NC State University, United States
REVIEW RETURNED	29-Apr-2020

GENERAL COMMENTS	This is a very well written and researched article. I appreciated the work that you all have done on the review. A few very minor clarification points as you finalize the piece for potential publication:  1) Abstract - the results say "our modifiable" factors - what do you mean here? 2) Abstract - conclusion - the first sentence is a bit of a run-on and could be broken into two for clarity 3) It is still a bit unclear in the parental feeding practices section about how often things were reported and by whom. I would suggest another read-through to make sure that is clear throughout. The socio-cultural section is very clear re: who reported key themes and how often they did. You don't need to quantify this, but in some places it might be helpful to say things like over half, a majority, all, or just 1-2, for example. Instead of only using words like some, or common. Thank you for this piece - well done!
--

REVIEWER	MILKAH N WANJOHI AFRICAN POPULATION AND HEALTH RESEARCH CENTRE - KENYA
REVIEW RETURNED	04-May-2020

GENERAL COMMENTS	All the issues raised in the first review are well dressed. The paper now reads very well. I recommend that the paper is accepted for publication, upon addressing these minor edits On page 32 line 14 (change seen to seem), Label the figures clearly, use the same caption as used in the text for clarity e.g figure 1 : Connecting sub-themes related to Theme One: Modifiable Infant Feeding and Care Practice
---